



# Clear, transparent, and timely communication for fair authorship decisions: A practical guide

Shahzad Gani[1,2], Lukas Kohl[3,4], Rima Baalbaki[1], Federico Bianchi[1], Taina M. Ruuskanen[1], Olli-Pekka Siira[1], Pauli Paasonen[1], Hanna Vehkamäki[1]

[1]Institute for Atmospheric and Earth System Research/Physics, University of Helsinki, Helsinki, Finland
[2]Helsinki Institute of Sustainability Science, University of Helsinki, Helsinki, Finland
[3]Department of Agricultural Sciences, University of Helsinki, Helsinki, Finland
[4]Institute for Atmospheric and Earth System Research/Forest Sciences, University of Helsinki, Helsinki, Finland

*Correspondence to*: Shahzad Gani (shahzad.gani@helsinki.fi); Pauli Paasonen (pauli.paasonen@helsinki.fi)

**Abstract.** Authorship conflicts are a common occurrence in academic publishing, and they can have serious implications on the careers and well-being of the involved researchers, as well as the collective success of research organizations. In addition to not inviting relevant contributors to co-author a manuscript, the order of authors, as well as honorary, gift, and ghost authors are all widely recognized problems related to authorship. Unfair authorship practices disproportionately affect those lower in

the power hierarchies — early career researchers, women, researchers from the Global South, and other minoritized groups. Here we propose an approach to preparing author lists based on clear, transparent, and timely communication. This approach is aimed to minimize the potential for late-stage authorship conflicts during manuscript preparation by facilitating timely and transparent decisions on *potential co-authors* and their responsibilities. Furthermore, our approach can help avoid imbalances between contributions and credits in published manuscripts by recording planned and executed responsibilities. We present

authorship guidelines which also include a novel *authorship form*, along with the documentation of the formulation process for a multidisciplinary and interdisciplinary center with more than 250 researchers. Other research groups, departments, and centers can use or build on this template to design their own authorship guidelines as a practical way to promote fair authorship practices.



## 1 Introduction

Collaborative and interdisciplinary research has been on the rise over the last few decades, and has increased the number of occasions in which authorship has to be negotiated among collaborating researchers (Van Noorden, 2015; Szell et al., 2018). Authorship conflict can arise as a result of unethical conduct or simply due to different perceptions of what is considered fair authorship practices (Floyd et al., 1994; Smith et al., 2020). These differences in perceptions and practices can cause authorship conflicts, including disputes over who should be included in an author list and the authorship order (Abbott, 2002).


Authorship disagreements disproportionately and negatively affect those lower in the power hierarchies due to their gender, sexuality, cultural background, and more. These discriminations can be overt or subtle. While subtle discrimination is often difficult to detect and lower in intensity, it can be as harmful as overt discrimination (Jones et al., 2016). Furthermore, these biases are present and amplified at all stages of knowledge production, not the least because those in positions of power tend

to assume the universality of their lived experiences. Gender bias against women, for example, was found to be prevalent in mentoring, collaborations, and peer review (Moss-Racusin et al., 2012; Larivière et al., 2013; Helmer et al., 2017). Discrimination due to sexual orientation and gender is widespread in academia where, like in most spaces, hetero-cis-normativity — a system of norms, privilege, and oppression that places heterosexual cisgender people above all others — is prevalent (Boustani and Taylor, 2020; Worthen, 2016). Furthermore, racism is another important and often underplayed factor

in research institutions (Harper, 2012), and manifests in knowledge production and resources being advantageous to past and present colonizers (Musila, 2019). In order to strive for fairness, we have to consider the intersections of these various identities that lead to bias, discrimination, and oppression (Crenshaw, 1989).

While prejudice and internal biases affect unfair authorship, disagreement also arises from differences in conventions of

various disciplines (Abrams, 2010; Hyland, 1999). Authorship criteria can be inconsistent among various publishers which in turn can differ from the ethical guidelines of the researchers' institutes (Silva and Dobránszki, 2016). Even within the same discipline, there are a diversity of views on what constitutes fair and ethical authorship (Smith et al., 2020). Furthermore, many unethical practices such as honorary, gift, and ghost authorship are not commonly studied outside of the medical community (Baskin and Gross, 2011). Honorary or gift authorship is the practice of including an author who did not contribute to the work,

as a favor or to provide more credibility to the work by using the individual's scientific standing (Greenland and Fontanarosa, 2012). Ghost authorship, on the other hand, is when someone heavily involved with the work is not listed as an author or disclosed their involvement in the work, usually when a financial conflict of interest is involved (Glezerman and Grossman, 2018).

Additionally, unplanned and ad hoc authorship decisions over the course of the manuscript preparations can cause further imbalances between actual contributions and credit. It is not uncommon for authorship issues to be considered extremely late in the manuscript preparation process (e.g., a few weeks before intended submission to journal). These unplanned and ad hoc authorship decisions may lead to late-stage authorship conflicts when the author list is finally being decided. It is even possible that an article ends up published without fair credit to those who contributed to it, leading to conflict and a toxic acceptance of

unfairness within the research community. Therefore, it is unsurprising that many studies and scholars have highlighted the importance of timely communication in avoiding authorship-related conflict and promoting fair authorship (Washburn, 2008; Dance, 2012; Zutshi et al., 2012; Hundley et al., 2013; Pain, 2021; Fleming, 2021). The Committee on Publication Ethics provides multiple resources for editors and publishers to facilitate ethical publishing, including those related to authorship (COPE Council, 2021). However, there is still a need for practical guidelines for individuals and institutes to think about

authorship in a systematic and timely manner.

Here, we present authorship guidelines developed for an interdisciplinary center as well as the process that led to the development of these guidelines. One of our key goals was to establish an institutional procedure to enhance the timely authorship planning, in order to minimize the potential for late-stage authorship conflicts during manuscript preparation and

to avoid the occurrence of imbalances between contributions and credits in published manuscripts. We use the principles of clear, timely, and transparent communication to formulate these authorship guidelines and a novel *authorship form* that can be used and adapted by other groups. We discuss the process of formulating these guidelines so that other individuals and





institutions can benefit from this experience. Formulating an institute's guidelines should be viewed as key part of committing the community to fair authorship practices and engaging individual researchers to think about the ethical dimensions of knowledge production.


## 2 Formulating authorship guidelines

The Institute of Atmospheric and Earth Systems Research, University of Helsinki (INAR) is a multidisciplinary and interdisciplinary center with more than 250 researchers. INAR researchers originate from a wide range of disciplines (physics, meteorology, forest sciences, chemistry, geosciences, environmental sciences, etc.) and career stages, as well as countries of

origin, and educational backgrounds. INAR has an Equality and Work Well-being Group (EWWG), operating since 2011. This EWWG consists of staff members ranging from professors to PhD students and a representative of the university administration and aims to ensure the well-being and fairness of the INAR community. Between Autumn 2018 and Autumn 2020, the EWWG was contacted multiple times related to unclear authorship practices, raising general issues rather than seeking support in individual cases. Thus, the group started to plan a process of collecting the concerns and developing helpful

authorship policies. Below we describe the process from identification of the problem to development and refinement of the guidelines. An overview of the process is illustrated in Figure 1.

To start the discussion related to authorship practices, the EWWG conducted a center-wide survey to identify specific authorship-related problems. The authorship issues experienced by the community were collected in a center-wide online

meeting on an online collaborative board (Flinga, https://flinga.fi/) which resulted in 7 broad categories (e.g., planning of co-authorship, co-authorship of data provider, authorship order, etc.). Participants of the meeting discussed authorship practices in smaller groups (Zoom breakout rooms) of approximately five people each, and it was possible to continue adding entries to the collaborative board even after the meeting. The EWWG distilled the collected experiences into a list of 22 items and conducted an anonymous survey amongst INAR staff on the frequency of encountering the listed issues (SI 3). Based on these

results, the group identified the five most common problems: 1) Author only comments on the final version of manuscript; 2) Author did not comment on any version of manuscript; 3) Disagreements on the order of author list; 4) Difficulty in bringing up authorship issues due to power imbalances; 5) Requests to add an author at a late stage. In a subsequent center-wide online meeting, participants were divided into five groups, each discussing one of these five problems in detail and suggesting practices to combat these issues. During this meeting, the groups were once mixed so that some participants could bring new

ideas to other groups, while some remained in the original group to tie the discussion with that of the original group. The comments from this meeting were again collected on an online collaborative board. Furthermore, eight volunteers from the center-wide meeting formed the authorship working group (AWG) with the objective of formulating the authorship guidelines based on these center-wide inputs and discussions.

The AWG met online three times and used the center-wide meeting feedback to prepare the authorship guidelines draft and a novel *authorship form* (Section 3.1 and SI 1). The draft of the authorship guidelines and the *authorship form* were circulated center-wide, acknowledging the issues addressed and those not addressed. Subsequently, a third center-wide discussion was organized in which INAR members were asked to provide feedback on the draft guidelines and form. This feedback was then used by the AWG to revise the guidelines and the *authorship form*. We formulated the idea of sharing the results in this

manuscript as we were close to finalizing the first version of the INAR authorship guidelines, and this manuscript became the first test case for using the *authorship form* in practice. We made a few minor edits to the form based on this experience and launched the authorship guidelines and form to the INAR community. We expect to further revise the *authorship form* and guidelines in 6 months based on feedback from the INAR community and envisage that there will be a need to return to the guidelines periodically (every few years) to account for evolving research methodologies and best practices in collaborative

research guidelines. The current version of the authorship guidelines and the *authorship form* are discussed in Section 3 and presented in full in SI 1. The *authorship form* of this manuscript also serves as an example use case (SI 2).





## 3 Authorship guidelines

While we discuss the process of preparing the authorship guidelines in the previous section, in this section we discuss the authorship guidelines and the novel *authorship form*, what problems they address, and their limitations.


We propose normalizing the idea of *potential co-authors*, acknowledging that authorship in a manuscript is fluid and that the final author list may not look the same as what was set out while conceptualizing the research or even start of the writing the manuscript. All collaborators on a manuscript should therefore be considered as *potential* co-authors until the finalization of the manuscript. A subset of these *potential co-authors* are the *main authors* — usually the first author and those who have

conceptualized the manuscript and are coordinating the work. The guideline proposes that if the *main authors* are early career researchers, this group should also include a senior researcher who supports the *main authors* in evaluating the co-authorship invitations and offers but does not necessarily become a co-author. As a practical tool, we propose an *authorship form* that helps with clear, timely, and transparent communication between *potential co-authors* around common authorship issues. We illustrate the use of the *authorship form* in Figure 2 (also see Appendix A).

### 3.1 Authorship form

The *main authors* initiate the *authorship form* when a manuscript is conceptualized and discuss their roles and responsibilities for the manuscript. We incorporate some of these early-stage decisions in the overview table of the *authorship form* (Table 1). It is at this early stage that we propose that the *main authors* tentatively decide the first author(s), the corresponding author(s), the last author, the method to decide the order of equally contributing authors (e.g., last name alphabetical or randomized), key

acknowledgements, etc. However, these decisions can be revised at any stage of the manuscript preparation by an agreement between the *main authors*. Furthermore, *main authors* can also be added (or removed) over the course of a manuscript preparation if needed.

The *main authors* create a *potential co-author* table (Table 2) for each *potential co-author* whom the *main authors* plan to

invite to contribute to the manuscript, which includes the past and planned responsibilities of this *potential co-author* related to the manuscript. The *main authors* should make the *authorship form* available to *potential co-authors* as soon as the discussion of the manuscript goes beyond the *main authors* and add any new *potential co-authors* as their role and possible contribution becomes clear.

Upon receiving the *authorship form*, all the *potential co-authors* agree on their assigned responsibilities and fill in the basic information (e.g., full name, affiliation(s), ORCID, etc.). This information is useful, for example, in the context of manuscript-related conference abstracts and presentations even before submitting the manuscript to a journal. The *potential co-authors* fill the rest of the entries in their table (checklist) in a timely way such that the *main authors* have all the information before submission of the manuscript to a journal. *Potential co-authorship*, as the name suggests, does not imply final co-authorship

in the submitted manuscript. The *main authors* use the completed *authorship form* to list authors, contributions, and acknowledgments in the manuscript submitted to the journal.

The *authorship form* should ideally be available as an online document to all *potential co-authors*. An online document gives everyone access to the most updated version of the form and enables them to see recent updates and who made them. For

manuscripts with many *potential co-authors*, having an online form also saves time for the *main authors* to get all the relevant information and share agreed tasks and responsibilities in a clear, timely, and transparent way. If the *authorship form* is not an online document, an up-to-date version of the document should be distributed among all *potential co-authors* upon request or upon major changes, to keep them updated with the tasks intended for them and the other *potential co-authors*.



### 3.2 Solving common authorship problems

In this section, we discuss the *authorship form* and the authorship guidelines in the context of common authorship-related problems. Some of these problems are authorship and authorship order, actual contribution to a manuscript, navigating power imbalances, and proper acknowledgments.

One of the most common problems causing authorship conflict is who gets to be the first, last, and/or corresponding author as
this decision often depends on highly subjective judgements which type of work (and whose work) is considered more valuable. In our proposed method, those authors who are most involved in the preparation of the manuscript — *main authors* — must make decisions on their roles early in the manuscript preparation and before getting heavily invested in the work. In case of random authorship order for all or part of the authors, we expect the *main authors* to decide the randomizing method (e.g., last name alphabetical) before sending the *authorship form* to rest of the *potential co-authors*. We also acknowledge that authorship
order and roles can be fluid and change over the manuscript preparation process. The *main authors* can make these changes to the *authorship form* at any time provided they reach consensus among themselves and communicate clearly with the other *potential co-authors*.

The criteria used to include someone as a co-author in a manuscript can often be unclear and inconsistent, and researchers
often have differing expectations based on their past disciplinary and institutional practices. In our method, the *main authors* list the other *potential co-authors* and their planned responsibilities early in the manuscript preparation process. *Potential co-authors* are explicitly invited to contribute to a manuscript, and others cannot expect to become co-authors without such an invitation. Ethics guidelines (e.g. COPE, etc.) and journal submission policies require that all *potential co-authors* must have read and agreed with the main conclusions of the manuscript to be included in the author list of the submitted manuscript. We
formalized this approval in the *authorship form*, where *potential co-authors* formally declare their agreement with the manuscript. We also clearly state honorary, gift, and ghost authorships as unethical practices in our authorship guidelines (SI 1). Furthermore, the *authorship form* requires proper attribution of contribution with the Contributor Roles Taxonomy (CRediT, https://casrai.org/credit/) and the use of the ORCID persistent digital identifier (https://orcid.org) which can promote integrity in scientific publications (Brand et al., 2015; Haak et al., 2012; McNutt et al., 2018). These contribution statements
lead to more accurate assessments, and all co-authors can receive adequate recognition (Sauermann and Haeussler, 2017).

The *authorship form* can also help lessen unfair authorship practices that result from power imbalances. For example, Trinkle et al. (2017) found that students were more likely to add faculty members as undeserving authors, especially if the faculty member was also their advisor. While our method does not provide a full solution for this problem, we still hope that an
emphasis on transparency and clarity on contributions can partially neutralize some of these power imbalances. In particular, we encourage use of the *authorship form* for assigning and recording contributions to the manuscript, which are transparent to all *potential co-authors*. We further suggest that the point of contact for all *potential co-authors* should not be the first author alone (often an early career researcher), but all the *main authors* and optionally also senior researcher (e.g., supervisor, group leader). This avoids that authorship decisions are made by a single early career researcher to a group of researchers more likely
to resist external pressure to add or remove co-authors. Furthermore, we stress that in cases when a very senior researcher is a co-author, they should not be the first, last, or corresponding author by default, but the decision should be based on actual contribution to the manuscript. For example, in cases when a relatively early career researcher contributes more to supervision of the students developing a manuscript, they should get appropriate credit in the authorship.

It is also sometimes unclear who is to be acknowledged and who is to be included as a co-author. Even though formal academic merits used for funding and position decisions are based on authorship, being acknowledged is a merit that should not be overlooked. We propose in the authorship guidelines that the people who made indirect or minor contributions to the manuscript should be acknowledged instead of being *potential co-authors*. However, we state that the decision of acknowledging a person instead of adding them as a *potential co-author* should be carefully discussed among the *main authors*.
Naturally, a *potential co-author* can choose to be acknowledged instead of being a co-author at any stage of the manuscript preparation, and we encourage this process of active self-evaluation. Beyond the authorship practices, we think that the





manuscript acknowledgements should also be considered as an indicator of scientific activity and the scientific community should appreciate its value, for example, in the curriculum vitae of the acknowledged individual.

It is important that the *authorship form* be read in the context of and as a part of the authorship guidelines (Appendix B and SI 1). Furthermore, while receptive to all inputs from the community, the AWG prioritized keeping the process simple, tangible, practical, and focus on center-wide consensus building. We wrote these guidelines to keep them applicable across a center that publishes hundreds of manuscripts in and across different disciplines with different authorship practices. The *authorship form* and the guidelines discussed here focus on improving communication and clarity around issues of authorship and do not

provide all answers related to authorship by themselves. For example, we could not define the exact criteria for being a *main author* or even a *potential co-author*, and how first, last, and corresponding authors should be decided. We leave these decisions to the *main authors* and hope that the emphasis on early communication helps avoid late-stage conflict. While most suggestions from the community made their way into the authorship guidelines, some were clearly recorded for future iterations of authorship or other center-wide guidelines.

**4 Lessons from the process of developing authorship guidelines**

The process of planning and formulating the authorship guidelines and the *authorship form* progressed steadily and was completed in a relatively short time (approximately 6 months; see timeline in Fig. 1). However, our work rests upon the EWWG's decade-long track record of contributing to equality and well-being in the community, which has helped leverage the authorship guideline process. The EWWG facilitated the identification of specific authorship issues within the community

and organized surveys and center-wide meetings to discuss the issues in detail. Furthermore, the formation of the AWG resulted in assigning specific people with the responsibility to turn ideas from the community into practical guidelines. It helped that all the center-wide and AWG meetings had clear agendas and the notes were shared with the center-wide community for accountability. The center-wide community was interested in these issues and provided invaluable feedback at multiple stages of the process.


The online and in-person dynamics of these discussions play an important role in voicing opinions, disagreement, criticism, and generally building consensus. Due to the ongoing COVID19 pandemic, all the discussions during the development of our authorship guidelines (EWWG, center-wide, and AWG) were conducted through video calls which may have played a role in facilitating open authorship-related discussions. For example, it may have been more intimidating to come into a physical

room and voice dissatisfaction about existing authorship practices. In-person meetings can be less inclusive, have more interruptions, and can cause higher anxiety (Sellen, 1995). It is also important to consider both synchronous and asynchronous forms of communication. While synchronous communications remove the factor of delay, asynchronous communications allow for flexibility (Robinson et al., 2017). In our case, having both asynchronous communication (online surveys) and synchronous communication (video calls) may have helped balance these communication aspects. Furthermore, the

collaborative board (Flinga) was used both synchronously (during Zoom meetings) and asynchronously (kept open for additional contributions after the meetings). Feedback from the community underlined the importance of the various communication channels. The center-wide meetings provided the opportunity to discuss authorship issues in a multiple heterogeneous groups and contexts. The online collaborative board created semi-anonymity, lowering the threshold for recording grievances, while avoiding personal attacks.


The *authorship form* and guidelines presented here are designed to be applicable across a diverse set of research disciplines and even for manuscripts with *potential co-authors* from multiple institutes. Furthermore, different groups can build their authorship guidelines and defined requirements on top of what we have provided here. Overall, based on center-wide feedback, we found that the concept of a *potential co-authorship* was welcomed by the community and the authorship guidelines and the

*authorship form* were expected to help in the preparation of transparent and fair author lists. Additionally, the community agreed that being listed in the acknowledgements of publications was also a valid indicator of scientific activity and should be considered valuable to curriculum vitae of the acknowledged individual. It is important to recognize that complete fairness in





authorship decisions requires striving for incremental improvements through continuous engagement and improving awareness.

## 5 Final thoughts

Academic publishing is essential for research careers and affects the professional and personal lives of researchers. Unfair authorship practices disproportionately impact on those lower in the current power hierarchies. Here we describe the process of formulating the authorship guidelines for a multidisciplinary and interdisciplinary center that publishes hundreds of papers every year. We report how this process led to a novel *authorship form* designed to help researchers discuss and decide authorship issues early in the manuscript preparation process at our institute. We hope that other groups can use our guidelines and the process we described to build their own guidelines. We provide a free license for other institutes to use and build on the method described and the form presented in this article. We can improve fairness in authorship practices by emphasizing on clear, transparent, and timely communication among *potential co-authors*. However, ultimately, fair authorship will eventually require recognizing, understanding, and tackling the role of patriarchy, racism, colonization, and intersectional issues in society and, by extension, in knowledge production.

**Data availability.** No primary data sets were used in producing this article.

**Author contributions.** SG, LK, PP, and HV are *main authors* as defined above; all other authors are listed in alphabetical order in the authorship list. Conceptualization: SG, LK, PP, and HV; Project administration: SG and PP; Methodology: All authors; Supervision: PP and HV; Visualization: RB, SG, and LK; Writing – original draft: SG and PP; Writing – review & editing: All authors.

**Competing interests.** The authors declare that they have no conflict of interest.

**Acknowledgements.** We acknowledge Teemu Hölttä for participating in the development of the methodology discussed here. The Equality and Work Well-Being Group (EWWG) of the Institute for Atmospheric and Earth System Research (INAR) initiated the discussion on fair authorship practices, the INAR community provided feedback through multiple centre-wide surveys and online meetings, and the INAR Board supported the process.

**Financial support.** This work is supported by the Academy of Finland ACCC flagship (grant no. 337549). Authors are supported by HELSUS Societal Impact Funding, Academy of Finland (grant no. 311932, 315204), European Research Council (grant no. 850614-CHAPAs, 692891-DAMOCLES), H2020 MarieSkłodowska-Curie Actions (grant no. 843511), and EMME-CARE (funding from H2020 Research and Innovation Programme (grant no. 697 856612) and the Cyprus Government).



**Appendix A.** Timeline and instructions for using the authorship form

- A *main author* initiates a new form by duplicating the authorship form and guidelines document — preferably as an online document. If using an offline document, all *potential co-authors* should be able to access the most updated version on requesting the *main author(s)*.
- The form should be made available to *potential co-authors* before or when the discussion of the manuscript goes beyond the *main author(s)*. Any new *potential co-authors* should be added as their role becomes clear.
- *Potential co-authors* fill basic information upon receiving the *authorship form* and the rest of the entries in their table (checklist) in a timely way such that the *main authors* have all the information before submission of the manuscript to a journal. Co-authors will be confirmed when the checklist is completed and reviewed by the *main author(s)*.
- In case of any queries or concerns, including addition of any name(s) not yet included, contact **all** *main authors*
  right away.

For details, see Figure 2 and the guidelines (Appendix B). The overview table and the *potential co-author* table of the *authorship form* are presented in Tables 1 and 2 (see SI 1.1 for the unabridged *authorship form*).

**Appendix B.** INAR Authorship guidelines

*Main authors*: The first author and those who have conceptualized the manuscript and are coordinating the work.

*Potential co-authors*: All co-authors are tentative till they have confirmed co-authorship, preferably with the authorship form. Keep in mind that the list and order of authors of each manuscript will be re-evaluated at the time of submission.

*Authorship form*: A form that helps with clear, timely, and transparent communication between *potential co-authors* around common authorship issues.

1) *Main author(s)* initiate the *authorship form* and share it with *potential co-authors* before or when the discussion of the
manuscript goes beyond the *main author(s)*. The *potential co-authors* need to complete the *authorship form* to become co-authors in the submitted manuscript. *Main author(s)* are the contact point of the manuscript, but if they are all PhD students or early-career post-docs, a senior researcher (e.g., supervisor, group leader) should also be included as an additional contact person. In such cases, the senior researcher is not necessarily a co-author, but should be included in all communications with the *main authors*.
2) *Main authors* decide *potential co-authors* by considering the people who made significant and direct contributions to the study or are expected to make such contributions by the time the manuscript is submitted. Co-authorship practices vary in different fields of science, so open discussion on expectations is advised. In most cases, a few authors write the first draft of a manuscript and in the case of several co-authors all don't have a major contribution to writing of the first draft of the manuscript. All *potential co-authors should be informed about the overall content of the manuscript early on (in
the form)*.
3) People who made indirect or minor contributions to the manuscript should be acknowledged. The decision of acknowledging a person instead of adding as a potential co-author is to be carefully discussed among the *main authors* (and the senior contact person if all *main authors* are early career). Acknowledgments including funding, infrastructure, assistance with technical issues, discussions, etc., should also be listed in the *authorship form*.
4) Each manuscript should follow a transparent and internally consistent criteria for co-authorship vs. acknowledgement. Combat your internal biases to make a fair assessment of actual contributions.
5) *Potential co-authorship* does not imply final co-authorship in the submitted manuscript. This "*potential" nature of the co-authorship* should be indicated clearly, and the final co-author list should be decided before submission of manuscript to the journal based on the completed *authorship form*.



a)   *Potential co-author* invitation can be cancelled if decided by *main authors* before *potential co-author(s)* put in significant amount of work and should be communicated clearly. Cases could include branching of manuscript or change of direction, missing contribution, etc.

   b)   *However, main authors* should not cancel *potential co-author* invitation if contribution already made does not end up in the final manuscript. In such cases it should be up to the *potential co-author* to choose to continue as a co-

author (still requires completion of *authorship form and associated tasks*) or not.

   c)   Even the first, last, and corresponding authors can change over the course of the manuscript preparation if the *main authors* agree. These changes should be reflected in the *authorship form* as soon as possible.

6) In case of random authorship order (except first, perhaps second/last), decide randomizing method (e.g., "last name alphabetical" which has the advantage of clearly indicating that these people have equal contribution) as soon as

possible.

7) The last invited *potential co-author(s)* should be given at least 2 weeks' time to comment on the manuscript and complete the authorship form. Consider that *potential co-authors* may need more time due to general vacation time across countries and cultures, parental leave, on field campaigns without internet, etc.

8) Honorary, gift, and ghost authorships are all considered unethical practices. All *potential co-authors* need to provide

proper attribution of contributions in the *authorship form* using the CRediT taxonomy.

   a)   Honorary authorship is the practice of including an author who did not contribute to the work, to provide more credibility to the work by using the individual's scientific standing.

   b)   Gift authorship is the practice of including an author who did not contribute to the work, as a favour.

   c)   Ghost authorship is when someone heavily involved with the work is not listed as an author or their involvement in

the work is not disclosed, usually when a financial conflict of interest is involved.

9) *All potential co-authors* are encouraged to comment on the final manuscript. In addition to any other authorship criteria decided by the *main authors*, *potential co-authors must have read the manuscript and agreed with the main conclusions to be included in the author list of the submitted manuscript.*

***Clear and timely communication is the key!***

*Note: This is version 1.2 of the INAR Authorship Form and Guidelines (last updated: 21ˢᵗ June 2021).* The *authorship form* and guidelines can be freely shared and adapted upon citation of this article.



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



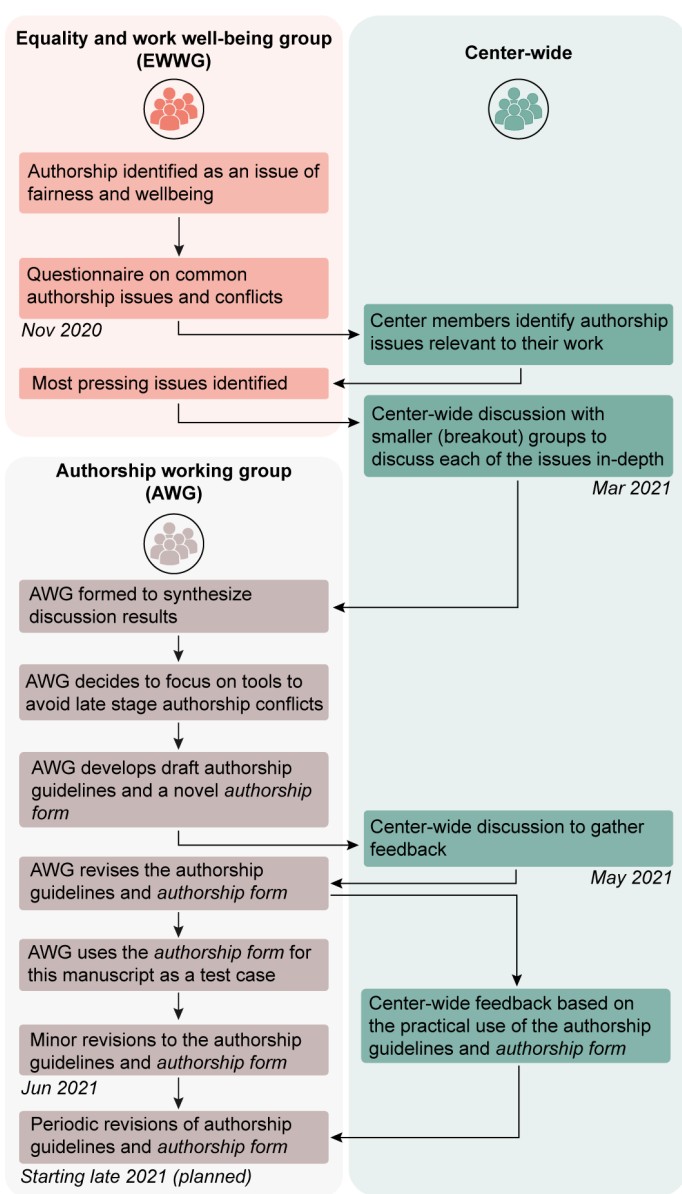

Figure 1: Flowchart describing the actors and actions leading up to the *authorship form* and guidelines.






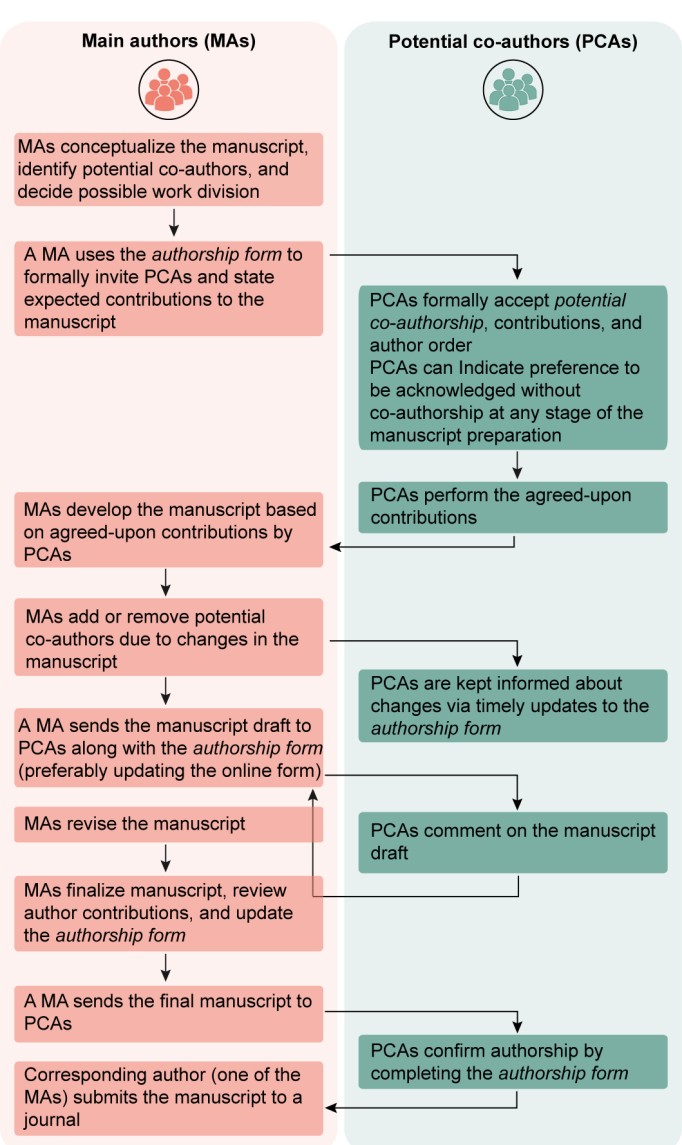

**Figure 2:** Flowchart describing the process of using the *authorship form*.




**Table 1:** The overview section of the authorship form. See SI 1 for unabridged *authorship form*.

| **Overview table** (Only for *main authors*) | |
|---|---|
| Manuscript (working) title/topic: | |
| Outline (2–3 bullet points): | |
| *Main author(s)* (all): | |
| First author(s) in order: | |
| Corresponding author(s): | |
| Last author: | |
| Co-authors with higher contribution, not in randomized order (in order): | |
| Randomization method for the order of equally contributing co-authors (E.g., Last-name alphabetical, coin-toss, a chess tournament, etc.): | |
| Leadership acknowledgement: | |
| Other people to acknowledge: | |
| Funding, projects, and infrastructure to acknowledge: | |





**Table 2:** Extract from the authorship form for each *potential co-author*. See SI 1 for unabridged *authorship form*.

| | |
|---|---|
| **Potential co-author table** (A copy of this table needs to be created for each *potential/main co-authors*) | |
| *Main authors* **create a table for each *potential co-author* and fill the first five entries below** | |
| Name: | |
| Date of invitation as potential co-author:<br><br>(Not for *main authors*) | dd.mm.yyyy |
| Date of removal from potential co-author: | dd.mm.yyyy |
| Past and planned responsibilities: | |
| Date(s) received manuscript draft(s): | dd.mm.yyyy |
| *Potential co-author* **whose name is listed above fills the following entries** | |
| *Fill the following sections upon receiving this form (Information provided can be used for manuscript-related conference presentations):* | |
| I agree to the planned responsibilities and accept *potential co-authorship* on: | dd.mm.yyyy |
| Full name for author list: | |
| Email: | |
| Affiliation(s): | |
| ORCID: | |
| Funding source(s) to be acknowledged (along with relevant details): | |
| *Fill the following sections any time before the manuscript is close to being submitted:* | |
| I want to be listed in the acknowledgment section of this manuscript, not as co-author<br><br>**(No need to fill rest of the entries if "Yes")** | Yes/No (dd.mm.yyyy) |
| Contribution to manuscript (E.g., using CRediT taxonomy)": | |
| Conflict(s) of interest (if any): | |
| I have read the manuscript and agree with the main conclusions: | Yes/No |
| Date of completion of checklist: | dd.mm.yyyy |