# Peer review of "Clear, transparent, and timely communication for fair authorship decisions: A practical guide"

_Geoscience Communication, 2021_

## Author Response (AR1)

Referee: 1

We want to thank you for your time in reviewing this manuscript and for the overall positive comments.

This paper describes a process - and the context in which it was carried out - to define a form to support transparency in authorship decisions. The nuance of the host and center dynamics is particularly well-articulated, showing the importance of a trusted body (in this case, the EWWG) in hosting difficult conversations. Also of general interest is the commentary on the impact of virtual workspaces on mediating open discussions and iterating toward a solution, particularly on enabling interactions among people located throughout power structures.

The workflow diagram clearly illustrates the iterative, reflective process and the interactions between the EWWG and the center-wide group. The authorship form is clear, as are the consistent and intentional use of terms: main authors and potential co-authors.

I would have liked to see a pilot of the form at the center, to demonstrate its utility in application. I'd like to learn how it is being adopted: are center researchers using the form because it works? Has the center decided to require or strongly encourage use of the form? The authors mention the broad research topics carried out in the center - how does this impact use of the form?

We agree that it would be useful to follow up on how the INAR community adopts the authorship form. We plan to conduct such a follow-up study and report on it in a future communication. However, we decided not to include such an analysis in the current manuscript due to three main reasons:

1) Timeline of the manuscript: the authorship form is designed to be used at an early stage of the manuscript preparation process (even though it can still be useful at a later stage as we have observed for some of our other in-prep manuscripts) and most manuscripts in our center take at least a year from conceptualization to submission. The inclusion of a meaningful pilot study in the current manuscript would therefore delay its publication by 1-2 years. Given the importance and timeliness of the issue of authorship conflicts, we gave preference to sharing our authorship guidelines (and the process of coming up with them) with the wider community before such a pilot can be completed.

2) Peer-review for improvements to the authorship form: We had planned to incorporate the external feedback (outside the center) through the peer-review as well as from other potential users. We wanted to harness this feedback (as we now have) and update the authorship form before we start pushing more strongly for widespread use within our center (version control can be a challenge once the authorship form is initiated for a manuscript). For example, while we had used version 1.2 of the authorship form for the

initial submitted draft of this manuscript, we will update it to version 1.3 (incorporating the mostly minor comments we have received) in the published version of this manuscript.

3) A peer-reviewed manuscript provides more (perceived) legitimacy to the entire process and enables us to illustrate the process and the thinking behind it to users in a more transparent and clear way.

All of that said, we know that INAR researchers have already started to use the form. We know this both anecdotally and based on some of us being main/potential co-authors in some of these manuscripts. The center strongly encourages the use of the authorship form in center-wide communications. To quote from the document itself: "This form is part of the INAR authorship guidelines and is recommended for manuscripts where all or some of the main author(s) are from INAR." We plan to assess the adoption and potential improvements to the form in an year or so.

The authorship form is specifically designed for broad research topics (even beyond science, as mentioned by the second referee). This means that researchers need to negotiate authorship across disciplinary traditions, which heightens the potential for conflict and requires that our guidelines allow for a relatively large degree of flexibility. Understanding the adoption of the authorship form across research themes is another area to assess in the coming time. Additionally, we are also keen on understanding how the authorship form will work in cases when some or most of the potential co-authors are from outside our center (most of the manuscripts from our center have one or more external co-authors).

Overall, your comments clearly indicate the next steps for the Authorship Working Group. These will require systematic surveys with the goal of understanding the adoption of the form (by research topic, career stage of main author(s), nature of collaborations, etc.) and potential improvements to it. We envision some INAR Authorship Working Group members performing this analysis and publishing a short piece on the findings in the coming year or two. Finally, we are also hopeful that other groups will experiment with/use the authorship form and share their own findings and experiences.

[Section 3.2 last paragraph] "It is important that the authorship form be read in the context of and as a part of the authorship guidelines (Appendix B and Section S1). Furthermore, while receptive to all inputs from the community, the AWG prioritized keeping the process simple, tangible, practical, and focused on center-wide consensus building. We wrote these guidelines to keep them applicable across a center that publishes hundreds of manuscripts in and across different disciplines with different authorship practices. The authorship form and the guidelines discussed here focus on improving communication and clarity around issues of authorship and do not provide all answers related to authorship by themselves. For example, we could not define the exact criteria for being a main author or even a potential co-author, and how first, last, and corresponding authors should be decided. We leave these decisions to the main authors and hope that the emphasis on early communication helps avoid late-stage conflict. While most suggestions from the community made their way into the authorship guidelines, some were clearly recorded for future iterations of authorship or other center-wide guidelines."

I would also have liked to see more guidance on contribution guidance. While it is important to be flexible to accomodate the broad research purview of the center, it would be helpful for adoption purposes to imbue more substance in the definitions of main and potential authors. The authors mention the CRediT taxonomy, but why not integrate this more deeply into the authorship defiinition process? Did the iiterative process find opportunities to improve the taxonomy?

We realized early in the process of hearing from the community that it would be extremely difficult to build consensus on and clearly define what constitutes main authors and potential co-authors. Especially since, to our knowledge, the idea of "potential" itself has not been used formally and intentionally in authorship discussions. By focusing on clear, timely, and transparent communication, we are trying to help the main authors think carefully, intentionally, and in a timely way about other potential co-authors, including who would be the main authors. That said, the main authors define past and present responsibilities for each potential co-author in the authorship form. Furthermore, completing the authorship form checklist requires each potential co-author to read the manuscript and agree with the key conclusions. While what we have right now is the bare minimum requirement for authorship, it may be possible for us (and/or other groups) to define clear criteria for main and potential co-authors in the future.

We used the CRediT taxonomy as an established tool pertaining to authorship and one that journals have increasingly started to formally require. We had not even considered improving on CRediT taxonomy in our surveys/discussions and at this stage, we do not have any suggestions on improving the CRediT taxonomy itself. In the "Contribution to manuscript" section of the authorship form, we do have a section for "Other" (in addition to the 14 current CRediT taxonomy categories). We/others can observe how the community uses this section to inform suggestions for improving the CRediT taxonomy in the future.

Overall, the paper is clearly written, describes a well-carried out research project, and shares a product that has the potential to advance and improve how the research community manages authorship. I can't wait to see how the authorship form is used.

We would like to thank you once again for your comments and feedback. We hope to incorporate many of your ideas in the future work of the Authorship Working Group.

Referee 2

We would like to thank you for your overall positive comments about the manuscript and the technical corrections which you have pointed out.

**General comments**

This paper has the potential to make a significant contribution to changing the way that authorship decisions are handled across disciplines throughout not just science but all of adademia. This process is suggested at the right time in our culture, when the conversation about increasing equity and inclusion is prominent in many institutions.

I like the way this paper is structured, using its own publishing process as a transparent test case to present how the co-authorship form would work. I can envision myself adopting this for future authorship discussions for my own publications. Thank you for doing this work and submitting it for publication!

**Specific comments**

I do not have any specific comments or concerns about the methodology or structure of this paper. I have a few technical corrections and suggestions, listed below.

**Technical corrections**

Throughout the document: It was unclear to me the reason for italicizing your key words. This seemed unnecessary.

We have now removed the italics throughout the manuscript and supplement (including tables and figures).

Line 81-82: I recommend splitting this into two sentences to improve clarity: "This EWWG aims to ensure the well-being and fairness of the INAR community. It consists of staff members. . . ."

Fixed

Line 90 et al.: Include date of last access for all Internet citations.

Included

Line 94: This is your first in-text reference to supplementary material. 1) It was unclear to me why your supplementary materials were labeled as "SI." 2) Consider re-ordering your supplements so that they are referred to in the text in the same order as they appears in the supplement.

We have now removed the term "SI". Instead, we now use (for example) "Section S1.2" (supplement defined upon first use). We can change further at the copy-editing stage depending on journal preference. We have also fixed the order of the sections in the supplement.

Line 99: Consider changing "once mixed" to "rearranged once."

Fixed

Lines 194-195: I had to read the sentence beginning, "This avoids that authorship. . . .," several times. It is awkward and would benefit from rewording.

[Updated text] "Including all main authors in authorship discussions and decisions will help avoid potential external pressure on a single early career researcher to add or remove co-authors."

Line 212: Change "focus" to "focused" for grammatical consistency with the other listed adjectives.

Fixed

Line 231: Change "play" to "played" so that it will be in past tense.

Fixed

Line 242: Delete the word "a" between "in" and "multiple".

Fixed

Line 257: Change "impact on those lower" to "impact those who are lower".

Fixed

Line 263: Delete "on".

Fixed

Line 263: Delete "However," and capitalize "Ultimately".

Fixed

Line 305: Change "till" to "until".

Fixed

Figure 1: I like this flowchart, but the way it is structured, I thought at first that the AWG was a subset or subordinant to the EWWG. Consider restructuring this flowchart so that the AWG sits in a third column.

Our initial draft of the manuscript had three columns which did not utilize the space well as you can see below (Figure R1). Therefore, we chose the more compact figure which we think is also more readable since text font size will be relatively larger in the figure that we submitted (it will be a single-column figure in the published manuscript).

[Figure]

*Figure R1: AWG separated in its own column.*

SI 1: In the Overview table, consider deleting "coin toss, a chess, tournament, etc." I feel like including these trivializes or makes light of the process. It is also inconsistent with every other mention of randomization method where only last-name alphabetical is mentioned.

Changed

[Updated text] "Randomization method for the order of equally contributing co-authors (E.g., Last-name alphabetical):"

SI 1: You defined the acronym for CRediT taxomony in section 3.2 of the main paper. However, SI 1 is referred to in section 2 of the paper, leading readers to perhaps encounter that term for the first time in SI 1. Please define that acronym upon first usage in the supplementary material as well as in the paper.

Fixed

SI 3: It might have been interesting to read the results of this survey in addition to seeing the survey form.

We do not provide these results due to ethical reasons related to data collection from surveys. Unfortunately, we did not envision a publication when we started this process (surveys/meetings) and therefore did not request permission from the participants to publish

their unedited responses. We therefore only provide the summarized responses in the supplement.

[Updated Section S1.2] "The five most common issues based on the center-wide survey addressed directly by the Authorship Working Group:

- Not commented on any manuscript version
- Only commented on the manuscript
- Disagreement concerning order of authors
- Power imbalance and personal preoccupation
- Someone asking to contribute to paper in last stages"

Editor:

I think the reviewers reports clearly outline the general usefulness of your guide, and are both very positive. Your responses are constructive and I can see from them that you have already implemented many of the minor revisions that were suggested: I am therefore more than happy to recommend publication subject to these minor revisions. I note that your responses to both reviewers (but particularly reviewer 1) indicate that you are planning further surveys on how the authorship guide is used, I think outlining some of these plans in the paper would be useful (there will undoubtedly be readers with similar thoughts that the reviewers have - outlining your plans can prime them for additional work around this?) - this could possibly fit in the "Final Thoughts" section, or as an additional short section after? Just something to consider.

Thank you for your suggestion. We have added some additional text in the manuscript to reflect your specific comments regarding future work.

[Section 4 last two paragraph]  "The authorship form and guidelines presented here are designed to be applicable across a diverse set of research disciplines and even for manuscripts with potential co-authors from multiple institutes. Furthermore, different groups can build their authorship guidelines and defined requirements on top of what we have provided here. Overall, based on center-wide feedback, we found that the concept of a potential co-authorship was welcomed by the community and the authorship guidelines and the authorship form were expected to help in the preparation of transparent and fair author lists. Additionally, the community agreed that being listed in the acknowledgements of publications was also a valid indicator of scientific activity and should be considered valuable to curriculum vitae of the acknowledged individual.

Assessing the usefulness and scope for improvements in the authorship form will require systematic survey of authors who have used the authorship form for preparing their manuscripts. These findings will likely depend on the research topics, career stage of main authors, nature of collaborations, and many other factors. Given the timeline from conceptualization of a manuscript to its publication, we have not yet conducted surveys around adoption of the authorship form. The AWG plans to conduct such surveys in the next year or two when substantial number of manuscripts will be completed using the authorship form from the conceptualization stage of the manuscript. We also encourage other groups that use this or a similar authorship forms to share their findings as well. It is important to recognize that complete fairness in authorship decisions requires striving for incremental improvements through continuous engagement and improving awareness."

Many thanks for your positive and constructive approach to responding to the reviews, and the review process in general. I look forward to seeing the amended manuscript.

Thank you for your comments and for overseeing the review process of this manuscript.